# Professional Identity Formation in Health Promotion Practitioners: Students’ Perspectives during an Undergraduate Program in Switzerland

**DOI:** 10.3390/ijerph182010754

**Published:** 2021-10-13

**Authors:** Verena Biehl, Frank Wieber, Denise Abegglen, Andrea Glässel

**Affiliations:** 1Institute of Health Sciences, Zurich University of Applied Sciences, 8401 Winterthur, Switzerland; frank.wieber@zhaw.ch (F.W.); denise.abegglen@zhaw.ch (D.A.); andrea.glaessel@zhaw.ch (A.G.); 2Faculty for Health Sciences, School of Public Health, Bielefeld University, 33615 Bielefeld, Germany; 3Department of Psychology, University of Konstanz, 78464 Konstanz, Germany; 4Institute of Biomedical Ethics and History of Medicine, University of Zurich, 8006 Zurich, Switzerland

**Keywords:** health promotion, workforce, undergraduate students, professional identity, focus groups

## Abstract

The health promotion (HP) community advocates for capacity building, quality assurance and political awareness of HP. Professional identity (PI) is of great relevance to these goals as persons who strongly identify with their profession better adopt their professional role, raising the quality, competence and common values within a professional group. However, investigations on the HP workforce are missing. In order to investigate PI formation in HP professionals, a longitudinal study was conducted with two student cohorts of a Swiss HP and prevention undergraduate program. Using a qualitative approach, focus groups were conducted at the beginning and end of the undergraduate program. Data were transcribed verbatim and condensed using thematic analysis. The results highlight the complexity of the HP’s professional profile. While students experienced difficulties to capture the profile at the beginning of the program, at the end they developed an understanding of it. The practical experience within work placements helped students to grasp the profile and specify their future professional role. Several behavioral, cognitive and motivational aspects were identified that influence HP students’ PI formation and can be fostered. For instance, universities can commit to public relations for HP practitioners and support the PI formation throughout the study program.

## 1. Introduction

In today’s society, one’s profession is becoming increasingly important and is seen as a central identity-forming factor in life [1,2]. Professional identity (PI) is understood as a complex construct of personal and social identity and relates to attitudes, behaviors, ethical values, commitment, quality awareness and competencies of the professional [1,3,4,5,6]. The formation of PI leads to the successful adoption of a professional role. If we apply this knowledge to the health promotion (HP) community it is worth investigating the PI formation for a better understanding of their challenges with regard to adopting the professional role and thereby strengthen HP in practice, research and policy. 

### 1.1. Professional Identity Formation

Over the lifetime, multiple identities of an individual co-exist, e.g., family, nationality, gender, and religion, and are rated differently depending on the individuals’ contexts [2,7]. It is assumed that the PI has a major impact on identity of an individual strongly influencing meaningfulness and structure of individuals’ lives [2,8]. Formation of a PI mainly happens in the transition from adolescence to adulthood and is, therefore, of great relevance to students [9,10,11,12]. PI is considered to be an indicator of various positive outcomes for professional and academic performance, such as strengthened social support, mental health (e.g., self-efficacy and self-worth) and life satisfaction [2,13,14,15,16]. Overall, common sense about (professional) identity is that individuals tend to think and act in line with the (professional) group, which is expressed by normative behavior and loyalty to the (professional) group [17,18,19]. A large body of research is based on social and developmental psychological identity theories such as the Social Identity Theory [13,18,20] and the identity-status model [21], which builds on Erikson’s work on identity development [22]. A main finding of these psychological theories is that identity, as well as PI, is built on personal factors and social factors, which obviously change over time. Building upon these theories, Mancini et al. (2015) explicated five dimensions of PI formation referring to motivational and cognitive dimensions as personal dimensions: (1) identification with commitment, (2) in-depth exploration, (3) reconsideration of commitment; a social dimension: (4) affirmation, and a behavioral dimension: (5) practices [16]. This theoretical background allows for the identification of relevant dimensions of PI formation in HP students, which forms the theoretical framework of this study. 

### 1.2. Professionalization of HP 

HP is seen as field of action of public health (PH) with greatest practical relevance and therefore, the need for special competencies in HP [23]. PH deals with a great variety of issues—including epidemiology, health care and health services research [23,24] making PH very broad and complex to capture [24,25,26,27,28]. Moreover, PH is described as research orientated [26,29], and still overlaid by biomedical principles [30,31] whereas HP is clearly practically orientated to address the social-environmental determinants of health at a community level. The principles, values and action areas of HP explicated in the Ottawa Charta [32] revolutionized PH over the last decades [33]. There are essential indicators for professionalization of HP, which claim for specific skills for HP within PH [34,35]. The driving force in this professionalization process is the International Union for Health Promotion and Education (IUHPE) promoting the dissemination of evidence-based knowledge to the HP community and advocating for HP [35,36]. The HP community has made great achievements, e.g., their own concepts and values, specific competency framework (CompHP) [37], university programs, handbooks, journals, conferences, an expanding accreditation system for HP practitioners and programs and a general implementation of HP in political health agendas (e.g., SDGs, national laws on HP and prevention) [38,39,40,41]. Besides these achievements and enablers of professionalization of HP some barriers have been recognized: a lack of institutional structures in practice and policy, no sustainable financing, competing interests in the health sector and beyond, a lack of visibility of HP or the complexity of the HP conceptualization [30,31,32]. Yet, the professional profile of HP practitioners is not well conceptualized, which is underlined by a still missing code of conduct of this profession. The professional profile is the basis for a PI formation. Educational institutions play a significant role for the PI formation of the future workforce. Practical experiences in the profession are essential for PI formation in educational programs [9,12]. More established professions investigate the PI formation of their workforce, such as medicine [42,43,44], physiotherapists [45,46] or social work [47,48,49,50]. To the authors’ knowledge, literature on the PI of the HP workforce is scarce, especially there is a lack of empirical studies [51,52,53]. So far, studies on professionalization mainly focus on PH or health sciences [27,29,54,55]. This study addresses this gap in the literature by investigating the PI formation of undergraduate students in HP and prevention at a Swiss university. 

### 1.3. The HP Workforce in Switzerland 

HP in Switzerland is quite well institutionalized, primarily with the federal foundation “Health Promotion Switzerland”. Even though, there are only few studies on the HP workforce in Switzerland. About 10,000 persons work in the field of PH, 40% of them in HP and prevention, without calculating HP at the workplace setting. Only 1/3 completed a special education in PH or HP and therefore a majority are lateral entrants to this field of action [56,57]. There is a lack of a young professionally trained PH workforce in Switzerland, which includes HP [56,58]. In response to this lack, the first HP undergraduate program was established at the School of Health Professions at Zurich University of Applied Sciences (ZHAW) in 2016. A postgraduate program of HP is still missing in Switzerland. Adopting the international HP standards, the undergraduate program is based on the CompHP [37]. This competency framework is an essential development in the professionalization of HP, because it enables transparency, comparability and quality assurance of HP in practice, policy and training [37,59,60]. The establishment of the HP undergraduate program at the ZHAW is an important milestone in the professionalization of HP and PH in Switzerland [61].

To ensure a successful start into the professional career of the future HP workforce, and therefore, a successful capacity building of the HP workforce, we investigated the PI formation of HP undergraduate students in Switzerland. This study, thus, aims to identify undergraduate students’ perspectives on their PI formation as future HP practitioners. 

The main research questions are:

1. What do HP undergraduate students associate with the professional profile of HP practitioners? 

2. How is the PI of HP undergraduate students formed during their undergraduate program?

3. Which promoting and inhibiting factors do the students indicate regarding their PI formation?

## 2. Methods

We applied a qualitative research approach to better understand the phenomenon and subjective experience of the study participants by using focus groups (FG) [62]. This exploratory, descriptive approach using thematic analysis [63] seemed appropriate as there is scarce empirical data on the PI formation of HP practitioners yet. As outlined in the introduction, we refer to Mancini’s et al. theory of PI formation [16], which was used as the underpinning theoretical framework for this analysis. Therefore, data analysis was primarily conducted deductively, followed by an inductive data analysis. We therefore explored the PI formation of the first and second cohort of HP undergraduate students at a Swiss University of Applied Sciences at the beginning and at the end of their study program. We followed the Journal Article Standards for Qualitative Research (JARS-Qual) to comply with scientific standards in qualitative research [64].

### 2.1. Study Participants and Relationship to Researcher

The researchers (VB and AG) conducted the FGs with the undergraduate students. Both are experienced FG moderators and have knowledge in HP in the Swiss context. Both are lecturing in the HP program and are thus familiar with the students’ world. This relationship may have influenced the participants’ statements but also allows for an atmosphere of confidence as the relationship between the researchers and students is considered to be open, trustful and positive. 

The study participants are quite heterogeneous regarding educational background and age. About 2/3 of the study participants do have prior professional training and work experience, mainly in the health sector (e.g., nursing, pharmaceutical assistants) or economic sector (e.g., business administration, retail). Other study participants directly enter the study program after graduating from high school. The 53 participants are quite homogeneous regarding gender (47 female), place of residence (German speaking part of Switzerland, mainly close to Zurich) and nationality (mainly Swiss, some Italian and German). For details see Table 1.

### 2.2. Participant Recruitment

All 1st semester students were invited to take part in the FGs discussing the PI formation as part of the regular lectures. This was the only inclusion criteria for study participation. Students were invited and informed orally during the lectures before the FGs and in written form. The same procedure was applied for participant recruitment in the 6th semester after collecting experience in the HP program and practice field. At both points in time, it was highlighted that study participation was voluntary and would have no influence on their academic evaluation. The recruitment strategy was convenience sampling trying to involve as many participants as possible at both points in time. Participants were randomly allocated to the FGs. Participation, at only one point in time, was also possible, and was not an exclusion criterion for study participation. Data collection was ended when saturation of relevant data on PI formation was reached so that further data collection would not generate new thematic categories. According to Guest et al. (2017), we assume a saturation of data within three to six FGs [65]. 

### 2.3. Data Collection

Applying standards for FG methodology allows a deep understanding and discussion of a collective phenomenon of a specific group. A homogeneous group composition promotes consolidation of a specific topic compared to heterogeneous group composition [62]. As outlined, FGs at the first point in time were conducted in the 1st semester of the two very first study cohorts (12/2016 and 12/2017). The exact point in time was three months after starting the study program to ensure rich discussion building on the first impressions of becoming a future HP practitioner. The second point in time of conducting FGs was in the 6th semester of the same two study cohorts (05/2019 and 05/2020) right before graduating. At this point in time students finished a work placement of at least six months, which promotes a rich discussion on PI formation as future HP practitioners. To sum up, we conducted five FGs at the first point in time containing 49 students, and six FGs with 53 students at the second point in time (see Table 1). This repeated measurement allows to investigate the PI formation over the course of the study program.

Data were collected in the context of the undergraduate program, which means the FGs were conducted at the university as part of the lectures. FGs in 05/2020 had to be conducted online due to the COVID-19 restrictions. Every FG was moderated by VB and AG and assisted by a second person to take field notes (DA, FW and others), and to record the discussions. All moderators and assistants were employed in the study program and specifically trained for this study by VB. Participants filled in an informed consent and a short survey on sociodemographic data before starting the FG. Sociodemographic data was collected on gender, age, nationality, place of residence before the study program and educational background (see Table 1). The duration of FGs was between 80 and 120 min depending on the engagement of the participants, which was in general higher at the second point in time.

Data were collected using a semi-structured interview guideline with open ended questions (see Figure 1), developed by VB and AG based on Mancini’s theory on PI formation for university students [16]. The guideline was pretested with colleagues at the university not involved in the HP undergraduate program (*n* = 4). Following their feedback, adaptations of the wording were made, and the guideline was finally structured in three parts: an introductory part, a main part that included theory-based core questions (see Figure 1), and an open end that enabled participants to add any comments or additional thoughts [62]. VB and AG prepared the FGs together, reflecting the questions and procedure before and after conducting the FGs. For better alignment of procedure and form of questioning AG joined VB at the first FG. FGs started with informing the students about the research project on PI formation, norms of discussion and on following the interview guideline. FGs’ questions were presented to the participants visually on PowerPoint slides. Audio records were transcribed verbatim and completed by protocols from each FG.

### 2.4. Data Analysis

Thematic analysis was used to capture the phenomenon and subjective understanding of PI formation of HP undergraduate students. This exploratory approach allows identifying a broad spectrum and the understanding of most relevant themes to describe the investigated phenomenon independently from the form of qualitative data [63]. Braun and Clarke describe six steps, which were applied to analyze the data of this study: Phase (1) familiarizing oneself with the data; Phase (2) generating initial codes; Phase (3) searching for categories; Phase (4) reviewing potential categories; Phase (5) defining and naming categories; Phase (6) producing the report. Data analysis was a chronological iterative process, going through Phase 1–4 of the transcript of point in time one (December 2016), later adding the transcript of the second study cohort (December 2017), checking for additional codes and categories. The same chronological iterative process was conducted for the transcripts of point in time two (May 2019 and May 2020). After coding and identifying first categories of all transcripts, final categories were refined, defined and named. Data analysis was mainly driven by deductive principles, trying to categorize codes in accordance with the theoretical framework on PI formation. Additional categories were derived inductively from data. As a final step, we explored the whole material of each category for searching overall promoting and inhibiting factors of PI formation. 

VB transcribed all transcripts and conducted the data analysis. For quality assurance DA double-coded about 25% of data and applied the coding guideline of VB proving applicability. Peer coding was used to train DA for applying VB’s coding rules. VB reflected on the final coding guideline with DA and AG for comprehensibility and completeness bearing in mind the contextuality of the involved researchers. Unclear codes were discussed and clarified by the three researchers. For final validation of the condensed results a Swiss expert on the professionalization of HP was consulted. The software MAXQDA 2020 was used for managing data analysis. 

### 2.5. Ethical Approval

The study was conducted according to the guidelines of the Declaration of Helsinki and was approved by the Cantonal Ethics Committee of Zurich (Req-2016-00711, 6 December 2016). Participants were informed about the study project and signed informed consent prior to the FGs. All participants provided written informed consent.

## 3. Results

Applying deductive data analysis regarding the theoretical framework of Mancini et al. (2015), three major categories were retrieved as dimensions of PI formation: (1) behavior, (2) cognition and (3) motivation. One additional major category was derived from data by inductive analysis named “professional profile” as central building block for PI. It is assumed that the professional profile is the basis for PI formation. Students first have to capture the nature of the professional profile before being able to integrate it into their identity. Within these four categories both personal and social factors of PI formation were identified. Results are structured regarding the three research questions: (a) the professional profile of HP practitioners associated by students; (b) dimensions of PI formation describing the single categories, namely behavior, cognition and motivation; (c) identification of major promoting and inhibiting factors regarding the PI formation. At the same time these factors can be valued as main differences within students’ PI formation. These differences are reported within the sections of each category. Subcategories within the categories are underlined in the text. Table 2 encompasses all identified categories, including the subcategories and codes. Each code is marked by being a promoting or inhibiting PI formation (see Table 2). Codes marked with both promoting and inhibiting factors (±) indicate that there are different perspectives of students on these topics.

### 3.1. Results to Research Question 1 

#### Professional Profile

This category encompasses aspects students associate with the professional profile of HP practitioners, which is seen as the basis to develop a PI. Personal as well as social aspects were relevant for describing and reflecting the professional profile. Overall, insecurity regarding the professional profile was recognized at the first point in time and decreased at the second point in time of data collection. This insecurity raised either regarding the comprehensibility and complexity of the professional profile or regarding how the new establishment of the undergraduate program fits the theory-practice-transfer. Some students are not sure to what extent they work “hands on” with the target groups or rather conceptual and at the desk. *“I have no concrete idea about this profession.”*
*(FG 4, T1).* Differences on the imagination of the professional profile were identified regarding age and prior work experience. Students who had a better understanding of the professional profile from the beginning were able to reflect and anticipate possible challenges regarding the professional choice, e.g., low societal recognition, or insecurity regarding the job situation. These were mainly more matured students with prior work experience. The following section describes details concerning the professional profile of HP practitioners, which the students mentioned.

Core elements of HP were described referring to the Ottawa Charta including “health advocate”, “health equity”, “participation”, “empowerment”, “mediate”, “salutogenesis”, focus on structural prevention, interdisciplinarity and “Health in all Policies”. Most relevant and challenging was the holistic, generalist perspective of HP, which offers comprehensive professional opportunities but is also very complex and hard to capture as professional profile as a student. *“We are trained very broadly—in a generalist way, and I sometimes ask myself if this is enough for a specific field of action, e.g., in addiction prevention”. (FG 1, T1).* This holistic approach was easier to grasp for students with prior work experience

A main factor in establishing PI and a professional profile is the use of terminology, i.e., the use of a professional title. In German, graduates of this new HP program in Switzerland are called “Gesundheitsförderer*in”; this title sounds rather generic and unspecific, which splits students’ opinions. There were differentiated discussions about the professional title regarding the comprehensibility, length, the differentiation between HP and prevention (which is not part of the title) or job search for HP practitioners or HP in the context of public health. There are different views on this professional title. Although some call themselves “HP practitioners”, most students will not use this expression. They would rather focus on the discretion in their future jobs, such as “project manager”. *"Problematic to me is that/ before I could always say I study health promotion and prevention at the university. And now: what am I now? I would never call myself health promotion and prevention practitioner. I think the name is so stupid. I promote your health. It sounds so overwhelming*
*[…]. I can’t identify myself with this name, I would never call myself like this.” (F5, T2)*.

Students use a wide range of different ways to describe their professional profile to family and friends or in other contexts. Most students use well-known practical examples of HP in Switzerland, such as famous information campaigns, to illustrate the professional profile. More mature students were more often aware of these examples. *“Mostly I use well-known examples, e.g., the Love-Live-Campaign […] and describe it and say that we figuratively do marketing in the health sector.” (FG 4, T1)*.

At the second point in time students mainly referred to the work placement in the last year of the undergraduate program. These practical experiences made by themselves or their fellow students contribute to specifying the description of the profile to others, but also lead to confusion. This is especially the case because in the HP community they mainly meet colleagues with different educational backgrounds. In some cases, the work placement organizations do not have a focus on HP as taught in the program based on principles of the Ottawa Charta and on a salutogenetic perspective, which confuses the students: *“I was at an organization for addiction prevention. We focused on prevention. Even though prevention and health promotion can’t be separated clearly, I got the feeling that the focus was on reducing risk factors and health promotion has moved into the background. I even described my job profile differently there. And now I realize I have to rebalance this again.” (FG3, T2)*.

Professional responsibility in the sense of a jurisdictional claim of HP goes along with societal and professional recognition. There is great concern that HP is not recognized and valued as an asset of competence and professionality, which is necessary to the field of action, because so far mainly lateral entrants with different educational backgrounds work in the field (e.g., psychologists, social workers, etc.). A majority of students realize a certain competition between professions and fear they have to fight for their own and specific recognition in the field. Students with prior work experience were more self-confident regarding their professional future, to name one more difference between students. *“I think it’s also a barrier, that other professions are afraid of us/ that they won’t accept us. I clearly experienced that with my aunt, who is a physician. When she heard health promotion, she said that’s ridiculous. She as a physician knows that the most important thing is cure. There is no prevention and there will never be prevention.’ We are a new profession, and this can be seen as a threat. The new ones are not well informed, or they can even take the work away.” (FG6, T2)*.

Obviously, students experience great ambiguity of societal recognition regarding the professional profile of HP practitioners from family and friends as well as in the professional context. They link this to the low public awareness of the professional profile of HP practitioners in Switzerland. The reaction to the professional profile in the public varies: open-minded, interested and positive vs. scrutinizing the purpose and reacting negatively. Skeptical reactions regarding the professional profile inside the professional community, from health professions, physicians or within the PH community promote students’ insecurity and intimidation. Lay persons often associate HP with medical professions. This is also underlined by the fact that HP is associated with behavior change interventions in society and in professional contexts: *“I know I have to eat apples and quit smoking. You don’t have to study that.“ (FG4, T2).* Students encourage themselves referring to be patient while raising awareness of the “new” professional profile of HP practitioners. Most students see themselves as promoters and advocates for the societal and professional recognition of HP, which is seen as a big challenge. 

### 3.2. Results to Research Question 2

#### 3.2.1. Behavior

This category summarizes the behavioral dimension of PI formation, which includes the actions relevant to the professional group. This includes the students’ perspective and their perception of society’s expectation about HP practitioners referring to personal and social aspects of PI formation. Differences between students regarding their professional behavior as a dimension of PI formation revealed that having positive experience, a successful theory-practice transfer and positive role models during their work placements, helped students to reflect more comprehensively on their professional behavior. In addition to that, students who experienced belonging to a professional group either at conferences or internships were more committed to the professional choice. Again, students with prior work experience are more aware of what professional behavior means.

Overall, most students primarily in the first semester directly relate the professional behavior to their personal behavior, and they refer less to the professional context. It seems a big challenge for some students to identify the right balance between a healthy behavior in a professional context and their healthy behavior in daily life. They mostly think they want to act authentically as an HP practitioner, which means they have to maintain a healthy behavior in daily life especially in the field they will work in (e.g., no smoking in the context of addiction prevention). They are not sure to what extent they act as role models for society or target groups. *“If someone conducts a quit-smoking campaign, one should not go outside and smoke a cigarette. Then you are no longer authentic. I think being a role model is professional behavior and that’s what I expect from myself. Private life remains private, and you can separate it from the job. But in the job, you should not preach water while drinking wine.” (FG3, T1)*.

They also name some characteristics to describe themselves as HP practitioners: empathic, unbiased, tolerant, patient, open-minded, talkative, matured and care-for appearance. They assume these characteristics as relevant for the professional context of HP and in addition they refer to critical self-reflection as a main aspect in HP practice. 

Reflecting society’s perception of HP practitioners, most students clearly state the role model as the main aspect. If their personal behavior does not comply with professional advice on healthy behavior, they are not taken seriously. *“I often get asked in my organization how I behave in daily life if I stick to the rules of WHO. I often get asked these questions, if I stick to healthy behavior in daily life, because I preach and study it.” (FG1, T2)*.

As professional behavior, some indicate the importance of reading professional literature and joining HP-related conferences. Some students pointed out the need of a professional community and staying in touch with the graduates of the undergraduate program. They confirm that a strong HP community would support PI formation in a long-term. It might be relevant to find a professional association for HP in Switzerland. This reflection on a more abstract level was easier for mature students. *“I think it is also important how our study cohort will develop. I think if we stay in contact among health promotion practitioners and keep the exchange, go to conferences, we will further be sharpening our ideas.” (FG3, T2)*.

Additionally, students confirmed that the discussion between their peers about all aspects of PI formation helped them to reflect their professional profile and PI formation. *“Some things have now [in the FG] become clearer to me about the professional profile. I think it’s useful to talk about it so that we really know what we’ll be doing later. It helps to talk about the professional profile.” (FG5, T1)*.

At the second point in time of data collection students revealed more aspects regarding the professional behavior in practice relating to standards and quality of professional action. Depending on the work placement some students struggle with missing theoretical foundation, missing quality assurance and missing focus on HP principles based on the Ottawa Charta. They hope and trust in themselves to slowly improve the professionality in HP practice. *“If someone comes in from university […] then you are not very welcome. It’s not the favorite topic in practice if you want to apply scientific work.” (FG6, T2)*.

The students themselves as interns or as young professionals are highly challenged when trying to change the state of the art in practice or at least to advocate for evidence-based HP. *“You’ve already noticed that you reach your limits quite quickly—you come into a company where a structure prevails—we have a lot of ideas or change requests. And sometimes you have to look at how you place that, that you stay true to yourself—you’re an intern, you’re the smallest in the chain—but nevertheless, you have to find a way to place it.” (FG1, T2)*.

#### 3.2.2. Cognition

This category encompasses a wide range of cognitive aspects regarding the PI formation, considering personal aspects of PI formation. Identifying main differences between students regarding the cognitive dimension of their PI formation, we again see both prior work experience, especially in the health sector, and positive experiences during their work placements influencing this dimension. The meaningfulness of HP as a professional choice was higher to these students. Differences within the level of reflection on the professional commitment got obvious between students in the first and sixth semester, being sounder in the sixth semester. 

Students critically reflect on professional requirements and competencies of HP practitioners. Overall, they point out what is most relevant to them as future HP practitioners: *(1) Innovative, analytical, and connected thinking, (2) patiently raising the societal and political recognition of health and especially HP and, (3) improving the evidence-based practice in HP.*

In addition, ethical reflections on HP practice were an important aspect of the FGs. Students reflect on the potential of their future profession naming structural (e.g., sustainable funding for HP) and content-related aspects (e.g., raising health inequity with HP interventions when not reaching vulnerable groups). Thanks to the work placements students are given the chance to reflect on the professional requirements needed to apply a theory-practice transfer. Overall, most students report great applicability of the learning outcomes in their work placements, e.g., communication skills, project management, evaluation and scientific methods. *“I was thrown in at the deep end during the internship—a very big challenge and a lot of responsibility and at the beginning I was swimming and a bit lost, but it gave me a lot of self-confidence and self-assurance, because I realized that what I do really turns out good and I can put into practice what I have learned.” (FG1, T2).*

Students with negative experiences in their work placements who had no successful theory–practice–transfer were not sure about the professional requirements expected from them. These students struggle to see themselves as professionals, as they think their generic knowledge is not enough for a specific field of action of HP. They consider doing a master’s degree or a vocational training to obtain a more specific basis of knowledge. 

Students in the sixth semester reflected in a very differentiated way about their PI and the commitment to HP. In the first semester, students had difficulty describing and understanding the professional profile and therefore stated that their PI is not established yet. Whereas others indicate that their personal values match with HP and therefore foreclose their PI as HP practitioners. Again, more mature students would not commit themselves from the beginning of the program to the professional choice, instead reflected on it sounder over the study course. In the sixth semester, students were more aware of the professional profile reflected in the work placement. They state that they realized the formation of PI over the three years of the program: *“At the beginning of my studies, they*
*[HP and person**] were two different things for me. And in the meantime, I can also identify well with HP and so they have now somehow grown together. I’ve noticed that I’ve adopted certain behaviors and ways of thinking that I certainly didn’t have before my studies. I find it interesting to observe that.” (FG3, T2)*.

Most students think their PI formation is not completed yet and assume the future workplace to have a big influence. 

The societal recognition of HP and the supportive context positively influences the commitment to HP and therefore, PI formation. In the COVID-19 pandemic most students’ PI increased because of the rising societal recognition of this major public health issue.

The FG discussions revealed a range of aspects that influence the reconsideration of the professional choice. The overarching aspect clearly is the job situation of HP, which negatively impacts the PI formation. It triggers concerns, distrust in the university and a negative mood. *“You sometimes ask yourself or you are even worried a bit whether there will be a place where you fit in. Do they need you at all? ” (FG2, T1)*.

Further reasons for reconsiderations are negative experiences in the practice or in professional contexts. *“At a HP symposium we told everyone enthusiastically that we are students in health promotion and prevention—then we were looked at/ not exactly nasty/ but in the sense of: ‘Competitors, what do you want?’ I stood next to one for five minutes and asked questions: She looked at me and said she didn’t have time.” (FG1, T2)*.

#### 3.2.3. Motivation

The category “motivation” in addition to “cognition” refers to the commitment and reconsideration of the professional choice and is associated with personal factors of PI formation. Differences between students again were revealed between those having positive experience during their work placements and those with rather negative experience. Another difference was also the outlook after the undergraduate program. Having a job perspective or a concrete plan after the program, strengthened students’ motivation. 

There are different intrinsic factors which increase students’ commitment with HP, e.g., the social aspect of HP focusing on health equity and *“doing something good for society” (FG6, T2).* This aspect gives meaningfulness to the professional profile of HP. Especially for students with a medical educational background the fact of acting in a preventative and health promoting way is highly motivating. *“I previously worked in nursing, and I no longer wanted to work at the bedside, but before. I was in the lung department, where so many patients came from smoking. I thought, you have to be able to do something beforehand so that it doesn’t get to that point. […] And then I looked at it (study program) and saw: “health advocate" and that motivated me quite a bit to possibly be able to change something.”*
*(FG3, T1)*.

During the undergraduate program the motivation shifted a lot, and the work placement was a highlight after four theoretical semesters. Again, the kind of experience in the work placement influenced students’ motivation. Positive feedback from colleagues or the target groups increased the motivation a lot. Negative experiences e.g., the awareness that HP at the workplace is only seen as a “nice to have” or negative recognition of their study program in a professional context was very demotivating for students and increased a reconsideration of their professional choice. *“On the one hand, the work placement was motivating, but on the other hand, it was also extremely demotivating. […] I saw that I could apply a lot from my studies. Demotivating was that*
*[…] there were negative reactions to the study program, also from my boss, who said he would not hire anyone who had done this study program. He would need a master’s degree or more. Then I thought to myself/ yes, fine/ three years done and the people who could be your bosses are rather negative.” (FG4, T2)*.

The majority of students value the broad spectrum of HP working in different settings and combining different tasks, e.g., desk work and engaging with the target groups. The learning outcomes of the program are motivating and interesting to most students and are valued for being adaptable to other sectors as well, e.g., communication or project management. This is especially promising to find a job in any field.

### 3.3. Results to Research Question 3 

#### Promoting and Inhibiting Factors for PI Formation

Analyzing each category of PI formation, overall, four main factors were identified influencing the PI formation of HP students without a ranking order: (1) “new” professional profile of HP practitioner, (2) practical experience, (3) job situation in Switzerland, and (4) meaningfulness of HP. These factors are at the same time the main differences between students’ PI formation. Another major difference identified is the temporal dimension of PI formation which was notable comparing FG in the first and sixth semester. 

(1) The fact that so far, the program has not been established in Switzerland provokes insecurity among many students. Other students’ value this as an asset to create their professional freedom and therefore valued this very positive. This *“new” professional profile* became more concrete during the study course and especially during the work placements.

(2) The *practical experience* during the undergraduate program clarified the professional profile and impacted the assumptions about the professional behavior. Students at the second point in time were aware of what belonging to a professional group means. Being connected to a professional community is experienced as highly conducive to PI formation. *“Now having a look into practice, one realizes how the puzzle fits together, for what one needs the single puzzle pieces, why we learned all methods and theories. It all comes together now—in the abstractness, a framework slowly forms in which there is room for identity.” (FG4, T2)*.

Moreover, these practical experiences influenced the commitment with the profession when the experience was positive and matched the students’ personal values and expectations of the professional profile. If the opposite was the case, it inhibited the PI formation and led to a reconsideration of the professional choice. 

(3) Another factor which changed over time was the perception of the *job situation in the field of HP* in Switzerland. This was already anticipated in the beginning of the program but became more realistic at the end before graduating. Therefore, on the one hand for many students this inhibited the PI formation, but on the other hand some students confirmed the commitment to HP even if the job situation was not easy. Some students already had job agreements, which also promoted PI formation and the motivation of other students. 

(4) Overall, students value the meaningfulness of HP and doing something good for society, which is clearly conducive to their PI formation. *“It is very meaningful to me doing something good for society.*
*[…] it is motivating to improve health in specific vulnerable groups.” (FG3, T2)*.

Summarizing the results on PI formation of undergraduate students in HP, we developed a model shown in Figure 2. The arrow shape of the figure indicates the processual nature of PI formation over the study course and not being finished when graduating. The professional profile on the top of the dimensions of PI formation indicates that the conceptualization of the professional profile is a central building block for PI formation. Overarching promoting and inhibiting factors for PI formation are added between the dimensions of PI formation and the professional profile as they directly influence both aspects. Dimensions constituting PI formation are presented as the categories revealed in the data. Personal and social factors influencing PI formation are integral components within the dimension and are symbolized in the model by the gradient white/grey coloring of the dimensions. 

## 4. Discussion

This study aimed to identify students’ perspectives on the professional profile, their PI formation and promoting and inhibiting factors of their PI formation during an HP undergraduate program at a Swiss University of Applied Sciences. To our knowledge, it is the first study investigating undergraduate students’ perspectives on their PI formation in HP. This investigation is important to understand the challenges of this young HP workforce, to find possible solutions promoting their PI formation. A strong identified workforce is supposed to engage more in their professional field and therefore promote the quality assurance of HP (e.g., more evidence-based practice).

The professional profile of HP practitioners is central to PI formation as it is seen as the basis that PI can be built on. This category was inductively added to existing dimensions on PI formation (behavior, cognition and motivation) (see Table 2). The main differences between students’ PI formation, both promoting and inhibiting factors, were the “new” professional profile, practical experience, the job situation and the meaningfulness of HP. The underpinning theoretical framework of PI formation by Mancini et al. (2015) helped to identify and describe the major categories as dimensions of PI formation. In the data, we found that the social dimension of PI formation could not be extracted separately, instead personal and social dimensions were found within all categories.

Some aspects of the study results are focused and discussed in the following sections.

### 4.1. Complexity of the Professional Profile of HP Practitioners

The professional profile of HP practitioners is hard to capture for students of the Swiss undergraduate program. Core elements, such as the holistic perspective on health, focus on populations instead of individuals, interdisciplinarity and strategies such as “advocate”, “mediate” or “enable” make it a very complex professional profile, as recognized in literature [66,67,68]. This complex nature of HP is underlined by the fields of action proposed by the Ottawa Charta [32]: To gain Health in all Policies an intersectoral and interprofessional collaboration is necessary, which demonstrates that HP is not the scope of a single profession. HP as a PH revolution should initiate a paradigm shift in all sectors of health and beyond. Literature on the professionalization of HP suggests a multilayered approach to implement the paradigm shift of HP: e.g., HP as a subject in school, HP as vocational trainings complementing prior educational backgrounds, HP as specific undergraduate and postgraduate courses and HP as complementary postgraduate courses [34,69]. Of course, this complexity is not easy to grasp for undergraduate students and explains their confusion about professional responsibility and the scope of their future profession. Therefore, the university has to choose adequate didactic methods (e.g., explorative learning [70]) and room for reflections on the professional profile and future professional roles [71,72]. There is a clear process identifiable in the data pointing out that some students can handle this complexity, which becomes clearer over the course of the study program and is seen as an asset within the great variety of HP. Some students are overstrained by this complexity reflected on a personal and on a social level of PI formation. This result has been shown before in HP students [71,73]. The issue of complexity also seems to be relevant to other professional groups, such as PH [25,26,27,28,54] or social work [47,48,74] which surely overlap the fields of action of HP. This overlap was also discussed in the FGs. Students were insecure about the differentiation of these professional profiles regarding professional responsibility. 

### 4.2. Professional Title “Health Promotion Practitioner”

Some students are concerned about their professional title as it is not specific and sounds unattractive. The differentiation between health promotion and prevention in the professional title of “health promotion practitioner” is also seen as a challenge. They reflected on the term of PH comparing it to HP and see some advantages in referring to established terms such as PH. Other students indicate that they name themselves HP practitioners as it reflects best their perspectives, values and professional profile. This insecurity might also be an aspect of this very newly established undergraduate program in HP in Switzerland. Students in Switzerland currently are still confronted with low publicity and reservations towards the program and the professional profile. This strongly influences their PI formation, which in turn is built on social aspects shown in other studies before [1,12,16,75]. The concern about the professional title is significant to PI formation. Students are aware of the numerous terms used for health-related study programs, e.g., PH, health sciences, health management, health psychology or health communication. All study programs and health-related professions claim to train for HP or for being competent for HP practice. This raises two questions for HP: (1) what do we mean by HP and which competencies are necessary and (2) why do different professions or educational institutions educate for HP but do not call it HP? The first question could simply be answered by referring to the CompHP, which is central to describe HP and to agree on common competencies for HP practice [37,38]. The main challenge is to further promote this framework in the HP community and beyond to refer to a common understanding of HP practitioners as a basis for ethical values and competencies necessary working in HP [60,76,77]. The second question is more complex to answer but is also related to the first question of a common understanding of HP. If the HP community does not insist on the term of HP, there is a great chance to be mainstreamed by other professions and topics. This mainstreaming of HP is already of concern to the HP community internationally [38,78]. A theoretical examination of this central aspect of professionalization indicates a clear need to insist on the term of HP. Language is central to fulfill a societal function referring to Luhmann’s systems’ theory [79,80] and to build up cultural, social and economic capital [81]. Therefore, we have to promote the term health promotion at least in the HP and PH community. The HP and PH community needs to foster the understanding and relevance of this term to the related professions who tend to mainstream HP. These activities advocating for HP have to be supported by the workforce, educational institutions and HP institutions such as professional associations. 

### 4.3. Practical Experience Influencing PI Formation

The undergraduate program strongly focuses on a practical approach enabling students to gain an insight into practice from the beginning of the program and to graduate with a period of at least six months of work placement. Results highlight the work placement to be a main influence factor for PI formation, which is confirmed in literature [12,16]. This is specifically supportive in the case of HP undergraduate students [71]. The influence can either promote or inhibit PI formation as students experience a major gap in the practice-theory-transfer. This gap has to be supported during the work placement to discuss students’ insecurities and concerns. Identifying these gaps in practice often referring to a lack of evidence-based practice motivates students to strengthen HP practice and advocate for higher standards and quality of HP practice. This can be valued as a strong commitment to HP, and therefore, a growing PI in HP. The status quo of HP practice in Switzerland seems to be of great diversity regarding quality standards. This may be due to the lack of specifically trained HP workforce in Switzerland [56,58]. Most students are very satisfied with their work placements and the match between learning outcomes and applicability in practice. Others reveal major gaps in their learning outcomes and the requirements in practice. Mostly, this is the case when organizations strongly focus on individual behavior change interventions, which is not the focus of the undergraduate program, but there is a strong focus on socio-environmental factors of health. Further concerns were mentioned regarding the job situation in the field of HP. Some students obtain job contracts at the work placement, which has a promoting effect on PI formation also for their fellow students. Others fear bad job opportunities after graduating. These insecurities have been revealed in similar studies in the field of HP [82,83] and, also, PH [26,27]. Karg et al. (2020) suggest supporting and coaching the students during their program to build their specialist profile within the HP profession during the undergraduate program promoting their job opportunities. Surveys with graduates of HP undergraduate programs in Germany reveal good employability [84]. These graduate surveys are elementary for analyzing the job situation of the future HP workforce and the basis for supporting students in the transfer from student to professional. It is assumed that a strongly committed HP workforce advocates even more for HP on the political agenda. PI is correlated to professional commitment and loyalty to the professional group [2,10,16,52], which may promote capacity building of the HP community. The educational institutions are in charge of creating a supportive environment for PI formation of their students [9,12], which is strongly influenced by the social recognition including the job situation of HP practitioners. Overall, students state that HP is very meaningful to them and therefore their personal commitment to HP is very strong.

### 4.4. Limitations

The study was conducted with the very first cohorts of HP undergraduate students in Switzerland. Therefore, the results may be limited to the generalizability to subsequent study cohorts as these students are the pioneers of this program in Switzerland and might have different experiences. This has to be considered when interpreting the results. Future studies could explicitly look at the differences of PI formation between the study cohorts or comparing it to different countries with more established HP undergraduate programs. Moreover, PI formation is surely not completed when graduating from university. Instead, PI formation is strongly influenced by the later work environment. Therefore, it is recommended to conduct further research on PI formation of HP practitioners working in the field. Conducting FGs with undergraduate students, quality standards of qualitative data collection and analysis were applied [64]. Results of data analysis were not checked by the participants, which may reduce the trustworthiness of the study results. The researchers regularly engage with the students of this undergraduate program, which leads to regular discussions and reflection on their PI formation helping the researchers to contextualize the study results, but which also entails the risk of increased social desirability in students’ reports. To promote methodological integrity, an external Swiss expert of HP professionalization was involved in the data analysis to find consensus in the data analysis.

## 5. Conclusions

To conclude, several aspects were revealed to support undergraduate students’ PI formation during their study program. This process is bound to continue after entering the job market. The complexity of the professional profile of HP is not easy to teach and learn within an undergraduate program. The university has to choose adequate didactic methods. Therefore, the study program needs to have a clear HP profile such as the CompHP and the lecturers have to be trained very well in HP and in teaching HP to facilitate students’ PI formation [73]. Another aspect regarding the undergraduate program is the supervision of the work placements. Significantly influencing PI formation, work placements have to be accompanied intensively to coach students’ reflections on the theory-practice-transfer and promote the adoption of the professional role. These supervisions could also prepare students for a successful transition into the labor market [82]. Furthermore, an important factor for PI formation is the publicity of the professional profile of HP practitioners. This should be supported by the university as they are in charge for creating a supportive environment for PI formation of their students [9,12]. Moreover, the promotion of the CompHP and the term health promotion have to be further promoted among the relevant stakeholders of the PH workforce including practice, policy and training. 

This study is a first explorative insight into the PI formation of undergraduate students in HP and prevention in Switzerland. An additional quantitative study is currently conducted to investigate possible influencing factors on PI formation and to compare HP to physiotherapy students who are known to have a strong PI. We need to conduct further studies in different contexts, both with students, with graduates and with professionals already working in the field to better understand the mechanisms leading to a strong PI as HP practitioner. 

## Figures and Tables

**Figure 1 ijerph-18-10754-f001:**
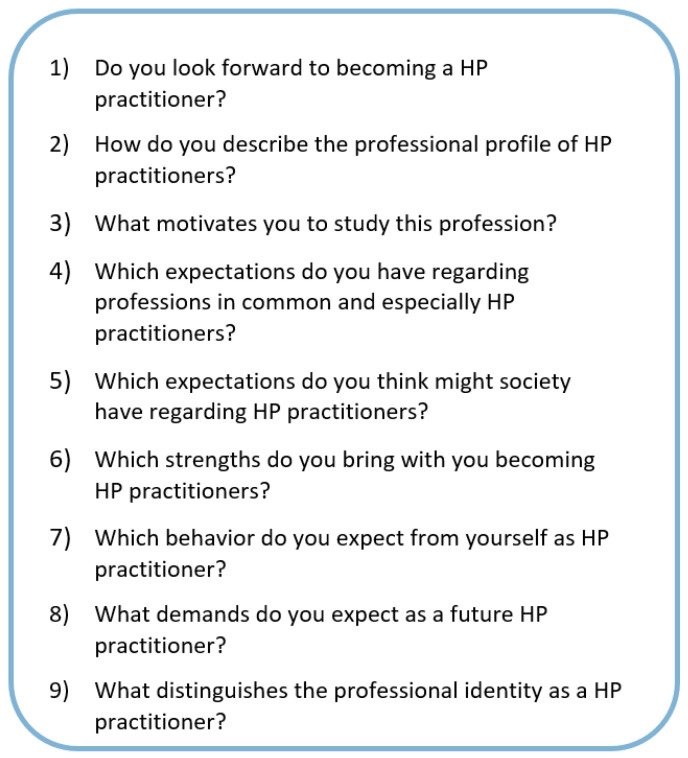
Interview guide of theory-based core questions for focus groups analyzing professional identity formation of undergraduate students in health promotion and prevention.

**Figure 2 ijerph-18-10754-f002:**
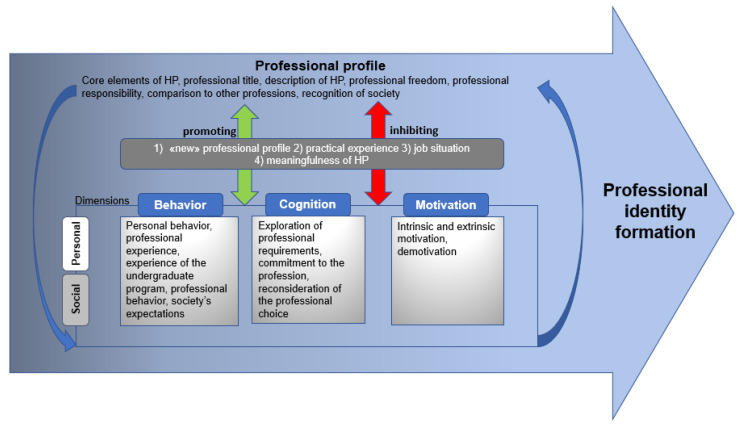
Model of professional identity formation of undergraduate students in health promotion and prevention in Switzerland derived out of the focus group results.

**Table 1 ijerph-18-10754-t001:** Sociodemographic data of focus group participants.

Point in Time of Data Collection	Study Cohort (Size)	Number of Participants per Focus Group	Nationality (Other Than Swiss)	Gender (Female)	Age Median (Min-Max)	Prior Vocational Training	Duration of Focus Group in Minutes
1st Semester (2016/17)	2016/19 (45)						
		9	2	6	24 (21–38)	6	80
		11	2	10	21 (19–30)	7	90
		12	0	11	23 (20–28)	8	80
		12	2	10	23 (20–36)	6	100
	2017/20 (35)						
		5	1	4	24 (22–29)	5	80
Total/Average		49	7	41	23 (19–36)	32	86
6th Semester (2019/20)	2016/19 (41)						
		8	1	6	25 (22–29)	4	120
		8	1	7	24 (23–30)	6	110
		9	2	9	24 (22–40)	7	100
	2017/20 (28)						
		10	1	9	25 (23–31)	8	110
		9	0	8	25 (23–29)	7	120
		9	0	8	25 (22–30)	6	120
Total/Average		53	5	47	25 (22–40)	38	113

**Table 2 ijerph-18-10754-t002:** Overview of categories, subcategories and exemplified codes indicating promoting or inhibiting factors for professional identity formation.

Categories	Subcategories	Exemplified codes Promoting (+) both (±) Inhibiting (−)
Professional profile		
(personal)	Core elements of HP	Health equity (+) Participation (+) Advocate (+) Mediate (±) Interdisciplinarity (±) Holistic perspective (±)
	Professional title	Mainstreaming (±) Comprehensibility (±) Specificity (±) Length (−) Attractivity (−)
	Description of HP	Personal interests (+) Modules in the undergraduate program (+) Well-known examples (+) Examples from practice (±) Complexity (−)
	Professional freedom	Flexibility (±) Forming the future of the professional profile (±) Autonomy (±)
(social)	Professional responsibility	Need of graduates of HP in practice (±) Competitional job situation (−)
	Comparison to other professions	Focus on health not ill-health (+) HP related to Social Work (±) Asset of competence of HP (±) Differentiation to vocational training (±)
	Recognition of society	Positive feedback (+) Negative feedback (−) Lack of publicity of HP (−) Comprehensibility (−) Association with medical professions (−)
Behavior		
(personal)	Personal behavior	Health behavior (±) Role model (±) Authenticity (+) Characteristics (+)
	Professional experience	Prior vocational training (+) Practical experience in work placements (±)
	Experience with the undergraduate program	Fit between learning outcomes and professional requirements (+) Holistic knowledge base (±) Methodological competencies (+)
(social)	Professional behavior	Belonging to a professional community (+) Professional association (+) Reading professional literature (+) Joining conferences (+)
	Society’s expectations	Role model (±) Educator for behavior change (−)
Cognition		
(personal)	Exploration of professional requirements	Reflection on professional roles (±) Reflection on principals of Ottawa Charta (±) Reflection on practice of HP (±) Theory-practice-transfer (±)
	Commitment to the profession	Match with personal values (+) Practical experience (±) Professional community (+) COVID-19 (±)
	Reconsideration of professional choice	Job situation (−) Questioning efficacy of HP interventions (−)
Motivation		
(personal)	Intrinsic motivation	Social aspect of HP (+) Meaningfulness (+) Acting before illness (+) Broad spectrum of work (+) Broad spectrum of settings (+)
	Extrinsic motivation	fair worktimes (+) fair wage (+)
	Demotivation	Theoretical semesters (±) HP only seen as nice to have (−) Negative feedback in the professional context (−) Mismatch of expectation and professional profile (−)

## Data Availability

The data are not publicly available due to restrictions regarding privacy and ethical consideration of the study participants.

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
