# Peer review of "Professional Identity Formation in Health Promotion Practitioners: Students’ Perspectives during an Undergraduate Program in Switzerland"

_ijerph, 2021, doi:10.3390/ijerph182010754_

Round 1
Reviewer 1 Report
Overall my impression is that this is a well conducted study, and the paper is nicely written. The findings were a little difficult to follow in one place - detailed below. I have reviewed the manuscript as a qualitative researcher without topic expertise, so my comments mainly relate to points of clarity, and qualitative methods/findings. Most are small. My 2 main comments to address are highlighted below with a *. I hope this feedback is helpful.
ABTRACT is well written
INTRODUCTION
- Line 31 typo “identi-ty”
- Line 39 I don’t know this topic but am surprised by the statement that professional identify has the biggest impact on an individual – 1981 is an old reference, do you have more recent evidence of this?
- Line 55 – do you mean little research done on the PI of the HP community? (if so this is also in lines 75-)
- In the UK health promotion units are now called public health units, and the term HP is quite outdated (this seems different in Switzerland?). Line 77 – if you are presenting HP as different to public health then I think early on you need to define both early on. This seems more important once the findings also allude to both terms.
- RQs are clear
METHODS
- Line 137 – purposive sampling on which characteristics?
- Line 170 – what is in Mancini’s framework? It would be good to know a little about this before it is used to organise your findings. Is it only behaviour/cognition and motivation in Table 2 that relate to the theory? or the categories/sub-categories?). Also how does it link to the SI theories you mention in the introduction?
- I think you need a sentence saying how many FGs at both time points and how many students took part once/both times (I know it is in the table, but a summary sentence would be great)
- Analysis - did you do any comparisons e.g. those who had prior professional training/work experience versus those who came direct from High school? Might that be interesting? *MAIN COMMENT
RESULTS
- There is a lot of good detail in here and careful analysis has clearly taken place. At times I found it a challenging read, perhaps because I am not familiar with the topic/underpinning theory. I had to re-read sections to follow them and be clear on how they differed to other sections.
- For me having the Table 2 and Figure 2 is overkill. They seem to include the same, with more detail in the Table?
- Lines 218-222 the text is more for the discussion than the results?
- You typically present the findings as a consensus view “students reflect”, students state” “students experience” etc. I would have expected some differences in views within/across the FGs – majority and minority views/negative cases. Did you look for these? (If so might they have related to the prior professional training/work experience versus those who came direct from High school?) *MAIN COMMENT
- Lines 319-326 – I am not sure if this is findings or discussion
- I’m slightly confused by the section “Promoting and inhibiting factors for PI formation”. It reads more as discussion? Also the other headings of this level clearly link to the 4 key parts of your table, this one does not.
DISCUSSION
- As a non-topic specialist reviewer, this discussion seems well written, with nice comparison to existing studies, thoughtful interpretation and good “so what” of the findings.
- I’d use your first paragraph, after the first sentence, to see how/why this study is novel and important. Then state the key take home findings more briefly (and does the theory you use hold up?).
- Line 652 – I’d avoid the word “biased” for qualitative research, consider saying something along the lines of the results are limited to ….we cannot claim these experiences would reflect those of subsequent cohorts
- Line 659 “Results were not reflected by…..” – what does this mean? Also again I’d avoid “validity” as its more a quantitative term – consider using trustworthiness
I rely on a section in this book to inform my thinking on these types of concepts in qualitative research: Ritchie, J., Lewis, J., McNaughton Nicholls, C., Ormston, R. (2014). Qualitative Research Practice. London, UK: SAGE. 2014.
Reviewer 2 Report
Thanks for the interesting manuscript. Some suggestions:
- Insert sub sections such as Background and Context into the current long introduction in order to enhance readability;
- The methodological framework is unclear e.g., What is a qualitative research design used in the manuscript? What is the analysis framework used in the manuscript?;
- The results are not obvious in answering the research questions;
- What are the significance/impacts of this study?
- Is there possible future study?
- It will be useful for the manuscript to go through professional editing and/or check through each sentence (there are some long sentences).
Round 2
Reviewer 2 Report
Thanks for the revised manuscript. It will be great if it could be professionally proof read at this stage in order to enhance the readability.